# Therapeutic Use of Bee Venom and Potential Applications in Veterinary Medicine

**DOI:** 10.3390/vetsci10020119

**Published:** 2023-02-04

**Authors:** Roberto Bava, Fabio Castagna, Vincenzo Musella, Carmine Lupia, Ernesto Palma, Domenico Britti

**Affiliations:** 1Department of Health Sciences, University of Catanzaro Magna Græcia, 88100 Catanzaro, Italy; 2Interdepartmental Center Veterinary Service for Human and Animal Health, CISVetSUA, University of Catanzaro Magna Græcia, 88100 Catanzaro, Italy; 3Mediterranean Ethnobotanical Conservatory, Sersale (CZ), 88054 Catanzaro, Italy; 4Nutramed S.c.a.r.l. Complesso Ninì Barbieri, Roccelletta di Borgia, 88021 Catanzaro, Italy; 5Department of Health Sciences, Institute of Research for Food Safety & Health (IRC-FISH), University of Catanzaro Magna Græcia, 88100 Catanzaro, Italy

**Keywords:** apitherapy, alternative medicine, bee venom, antioxidant activity, antimicrobial and antiviral activity, anti-inflammatory activity, anti-cancer effects

## Abstract

**Simple Summary:**

Bee products consist of many substances that have long been known for their medicinal and health-beneficial properties. Venom is certainly the one that has attracted the most interest due to the complexity of its chemical composition. Several types of research have been conducted utilizing biological (cellular) systems to figure out the properties of bee venom in vitro. Primarily, cell lines of various sorts and origins are used for this purpose. Afterward, experiments on murine models paved the way for clinical trials on humans. Therefore, there are numerous reviews summarising the uses of venom for human medicine, but none have focused on its use in veterinary medicine. This review aims to gather the relevant publications on the use of bee venom in veterinary medicine.

**Abstract:**

Apitherapy is a branch of alternative medicine that consists of the treatment of diseases through products collected, processed, and secreted by bees, specifically pollen, propolis, honey, royal jelly, and bee venom. In traditional medicine, the virtues of honey and propolis have been well-known for centuries. The same, however, cannot be said for venom. The use of bee venom is particularly relevant for many therapeutic aspects. In recent decades, scientific studies have confirmed and enabled us to understand its properties. Bee venom has anti-inflammatory, antioxidant, central nervous system inhibiting, radioprotective, antibacterial, antiviral, and antifungal properties, among others. Numerous studies have often been summarised in reviews of the scientific literature that have focused on the results obtained with mouse models and their subsequent transposition to the human patient. In contrast, few reviews of scientific work on the use of bee venom in veterinary medicine exist. This review aims to take stock of the research achievements in this particular discipline, with a view to a recapitulation and stabilisation in the different research fields.

## 1. Introduction

A third of our food is produced directly or indirectly by honeybees, one of the most abundant and effective pollinator species. Additionally, these insects are bred, creating worthwhile products and job prospects in rural regions [1,2,3]. Various bee products are used in alternative medicine, as they possess interesting healing and disease-preventive properties. The term “Apitherapy” is used to describe a group of therapeutic and preventative procedures used in both human and animal medicine to improve health [4,5].

The use of honey, pollen, and royal jelly spans the fields of nutrition and food sciences as well as pharmacology (desensitization, anti-inflammatory therapies, and treatments for autoimmune diseases).

The medicinal and nutritional benefits of hive products have recently been thoroughly studied. Numerous studies have been conducted, not least due to the rising problem of drug resistance, which has necessitated the search for novel pharmacologically active substances [6,7]. Additionally, in some parts of the world, drug expenses are prohibitive, whereas apitherapy cost is typically reasonable. Consequently, apitherapy has been used as a “supplemental medication” in many different nations (Brazil, China, Japan) [5]. However, there is no shared consensus on the medical application of apitherapy [5]. The methods of administration and use of hive products vary widely. Use must be based on an understanding of the unique physiological reactions of humans and animals, in addition to age and weight [8]. Because bee products are multi-substance mixes with several unknown components, they provide a challenge [9]. It is important to bear in mind that, in the absence of products with standardized chemical composition, the therapeutic effects of bee products may fluctuate according to the source. Soil, climate, harvesting and storage practices, botanical sources, etc., all affect the quality and effectiveness of the products used in apitherapy. To maximize outcomes and protect the patient from hazards, it is crucial to understand the conditions under which the products are generated. In any case, researching potential and standardizing products seem promising alternatives for the near future.

Honey, propolis, pollen, bee bread, royal jelly, apilarnil, beeswax, and venom are some of the bee products utilized in apitherapy [10]. Bee venom and its clinical veterinary medicinal uses are the main topics of this review. Bee venom’s medicinal benefits have a very long history. Since 3000 BC, bee venom has been utilized in traditional Eastern medicine to treat inflammatory illnesses [11]. In this context, ancient civilizations such as Nibia, Babylon, and Assyria would also have been familiar with bee venom. The Greek Hippocrates, who is considered the “father of medicine,” employed it to treat arthritis and other inflammatory conditions [10]. The Roman Pliny the Elder (23–79 AD) describes it in his *Naturalis Historia* [8,12]. Charlemagne is said to have used bee venom to cure his gout [13]. The first collecting attempts occurred in the 19th century. Between 1897 and 1899, J. Langer of the University of Prague made the first attempt to harvest the venom without killing the bee, collecting the poison in droplets inside capillary tubes. His first technique involved pressing a bee’s lower abdomen so that she would protrude a sting with a drop of poison at the end. In the early 1900s, apitherapy began to be used to treat rheumatic disorders. This practice quickly expanded throughout Europe. Demand for the product has become stronger. The poison was first made available for purchase in 1930 by the Mack company in southern Germany [14]. The first methods of manual extraction of the poison were followed by more articulate ones [15]. Using a light electric shock to induce the bees to inject the venom was a time-saving method. The drawback was that the bees lost their stinger and, in part, their abdominal guts, resulting in death. This method was perfected in Czechoslovakia in 1960 when the mean used for collection allowed the bees to stay alive and maintain their stinger [14]. Bee venom therapy was pioneered by Dr. Bodog Beck, a naturalized Hungarian-American who published the influential book, *Bee Venom Therapy*, in 1930. Dr. Joseph Broadman of New York began using bee therapy to treat arthritis in the early 1950s and wrote about his experiences in the 1962 book, *Bee Venom, the Natural Curative for Arthritis and Rheumatism* [11]. Since 1971, at least 12 nations in Europe, three in Asia, and three in America have used bee venom [14].

Bee venom has traditionally been used to treat inflammatory diseases such as rheumatism [16,17]. However, bee venom has also been known to be used as an adjunct in the treatment of neurological disorders, asthma, and infectious diseases such as malaria. There are limited research evaluations on the use of bee venom in veterinary medicine. In contrast, many published studies have addressed its potential application in human medicine. In light of the latter consideration, this study aims to summarise published research on the potential application of bee venom in animal therapies.

## 2. Venom Source

Bee venom is produced by the venom gland located in the abdominal cavity of female honeybees [4]. The gland is connected to a containment sac. The veliniferous apparatus of social insects belonging to the genus *Apis* is an essential defence mechanism. Bees sting in the vicinity of the apiary in an attitude of colony defence [18]. The queen, on the other hand, stings to kill rivals [19]. Each hive can only have one queen, and when several queens are born at once, either some of them escape along with a specific number of bees, a born queen kills the unborn queens who are still within their cell, or two queens engage in a death battle [20]. The protein concentration of queen bee venom is highest in the first (0–3) days of life and diminishes after 7 days (a necessary condition shortly after emergence to kill the eldest queen and twin queens in competition for hive domination). As the gland degenerates, the protein content of the venom in honeybees diminishes during the following days. In contrast, the venom is not detectable at the time of emergence in the female honeybees. Instead, it increases quickly over the following two days, remains constant for the first 14 days, and then drops. Therefore, older honeybees produce less poison than younger ones [18]. The venom’s composition changes throughout time with age. For instance, melittin is released in an inactive precursor form, which transforms into an active form with growth and the passage into the guardian stage, which happens about day 20 of age [18].

Honeybees have a pointed stinger that is extracted from the abdomen during stinging along with the venom sac. Unlike wasps and hornets, they can only sting once before dying [21]. When a bee stings a person or a mammal in general, the stinger remains embedded in the skin, and the bee dies as a result of ripping out its intestines, muscles, and nerve center in an effort to detach. The bee dies because such a large amount of its body is lost. The stinger’s pointed end features tiny hooks that keep it from being removed without damage. Once embedded, it uses a separate piston mechanism to push the venom into the wound [22]. The stinger self-incorporates into the tissue and there is a simultaneous release of the contents of the venom sac, which is usually expelled completely within a few minutes [21]. Additionally, the alarm pheromone message conveyed by bee venom activates other bees to defend the hive. The alarm pheromone is made up of the mandibular gland’s 2-heptanone molecule and other substances, such as isopentyl acetate, released by the gland connected to the stinging apparatus. [21,23]. Bee venom cause localized inflammation with symptoms like pain, heat, and itching to systemic allergic reactions that can end in anaphylactic shock and, in extreme cases of hypersensitivity, can be fatal. [24,25]. In popular culture, bee venom is frequently connected to these phenomena. However, it is one of the most priceless gifts the beehive has given us. It can be helpful in the treatment of a wide range of illnesses when used in tiny dosages. Its use in the treatment of many illnesses states is intriguing due to its complex composition of chemicals with significant pharmacological and biochemical activity.

Freshly secreted bee venom is a clear, colourless liquid that forms a light yellow powder when it dries [26]. It has the pungent aromatic odour of honey and is acidic (between 4.5 and 5.5). The water content in bee venom varies between 55 and 70 percent [27]. The active components of the venom of various hymenoptera are peptides, proteins, enzymes, low molecular weight substances, and aliphatic constituents in varying quantities [18,28]. No exemption applies to bee venom. It is a very complicated composition that is largely (80%) made up of proteins. These latter compounds have high (proteins) or low (peptide) molecular weights. Biogenic amines are the most important low-molecular-weight substances. The peptides in bee venom adolapine, melittin, apamin, and peptide 401 have undergone extensive research [29] Table 1 summarises the composition of bee venom.

The right collection is the crucial element in obtaining the best quality bee venom. Pollen, honey, and other colony products must be free of impurities. As pointed out by Krell [15], there are no official quality standards, as bee venom is not recognised as an official drug or foodstuff. A quantitative study of its more stable or readily quantifiable components, such as melittin, dopamine, histamine, noradrenaline, or those for which contamination is suspected, can be used to determine a substance’s degree of purity. Standardization and quality control techniques for the efficacy and purity of hymenoptera venom, particularly those of bees, were discussed by Guralnick et al. in 1986 [30].

## 3. Venom Constituents and Their Biological Activities

### 3.1. Melittin

The more abundant element is melittin. It makes up roughly 50% of the peptides in the venom and consequently 50% of the dry BV [31]. The sequencing of the peptide by Habermann and Jentsch (1967) revealed that it included 26 amino acid residues [32].

The carboxy-terminal region (residues 21–26) is hydrophilic due to the presence of a positively charged amino acid stretch, whereas the amino-terminal region (residues 1–20) is hydrophobic [33,34]. Melittin is soluble in water as a monomer or tetramer due to its amphoteric nature. The polypeptide spontaneously integrates into the phospholipid bilayer of cell membranes, damaging them [35,36,37]. Thus, the molecule’s primary structure resembles the fundamental form of a detergent-type molecule. By changing the phospholipid composition of the cell membrane, melittin accumulation leads to cell lysis. Neumann and Habermann (1953) discovered that it was a haemolytic factor for the first time [38]. Melittin is a member of the class of compounds known as amphiphilic due to its distinctive structure. Each melittin chain has two α-helical segments and resembles a bent rod overall. Melittin is monomeric at the lowest concentrations necessary for cell lysis and tetrameric at the amounts found in the bee venom sac [39]. To describe the precise steps involved in membrane permeation by amphipathic α-helical lytic peptides, two different pathways were put forth. They are theoretically quite different from one another.

The first one, known as the “barrel-stave” model, is characterized by the insertion of amphipathic α-helices into the membrane’s hydrophobic core to create transmembrane holes. In the second, known as the “carpet” model, the peptides are in touch with the lipid head group during the whole process of membrane permeation and do not integrate into the hydrophobic core of the membrane, even if they are not required to acquire an amphipathic α-helical structure [40]. The cytotoxic and anti-inflammatory actions of melittin on tumor cells may be significantly influenced by these structural characteristics. By facilitating an enhanced Ca^2+^ ion influx, melittin also activates phospholipase A2 and adenylate cyclase [41]. All of the above characteristics make melittin interesting due to its currently known anti-cancer, anti-inflammatory, antiviral, antibacterial, and neuroprotective properties [42,43,44,45].

### 3.2. Apamin

Eighteen amino acids make up the polypeptide apamin, which also has two disulfide bridges in it. It is the smallest neurotoxin in bee venom and makes up less than 2% of the weight of dry venom [46]. Apamin functions as an allosteric inhibitor since it has long been recognized as a highly selective blocker of small-conductance Ca^2+^ -activated K^+^ (SK) channels [47]. These channels help control the ionic balance in the cell membrane, which regulates the resting and action potentials in vital cells as well as signal transmission through neurons and muscle contraction. Through this route, apamin produces a neurotoxic impact by mediating long-term after-hyperpolarization in neurons and muscle cells. This polypeptide can also pass the blood–brain barrier and affect how the central nervous system functions through a variety of mechanisms. For instance, it has been shown in rats to have neurotoxic effects that result in hyperactivity and convulsions [48]. Apamin also affects the permeability of the cell membrane to potassium (K+) ions by inhibiting calcium-activated K+ channels. Through the Akt and Erk signaling pathways, the toxin can prevent vascular smooth muscle cells from proliferating and migrating [49]. This finding emphasizes apamin’s potential for use in atherosclerosis treatment plans. Generally speaking, pathophysiological responses, including atherosclerosis and Parkinson’s disease involve a significant function for apamin target channels.

### 3.3. Mast Cell Degranulation Peptide

It also is known as “peptide 401”. It is a polypeptide with 22 amino acid residues and a molecular structure that resembles an apamin with two disulfide bridges [50]. It only makes up a small portion of the venom, roughly 2–3% of the dry matter volume. The name MCD refers to the physiologic process by which mast cells release histamine; the peptide promotes mast cell degranulation and sets off inflammatory responses. In animal trials, MCD has been proven to significantly lower blood pressure [51]. In this context, it is the component considered responsible for the hypotension observed in BV intoxication [52].

### 3.4. Adolapin

It is a polypeptide of 103 amino acids and makes up around 1% of dry BV. By inhibiting prostaglandin synthesis and cyclooxygenase activity, adolapin exerts anti-inflammatory, analgesic, antinociceptive, and antipyretic effects [53,54]. Because naloxone has been shown to partially suppress the analgesic effect of adolapin, a central mechanism may be involved in the drug’s action [55]. According to Koburova (1985), adolapin, like aspirin and other comparable substances, has antipyretic actions, most likely via inhibiting the synthesis of cerebral prostaglandins [56].

### 3.5. Phospholipase A2

The enzyme that is most prevalent in bee venom is phospholipase A2. It is an alkaline component that makes up 12–15% of the dry BV. It has four disulfide bridges and 128 amino acid residues [57]. It is highly aggressive against the cell membrane, resulting in cytolysis, together with melittin and lysolecithin, which are produced when phospholipase acts. Furthermore, phospholipase is the most significant allergen and hence the most toxic element of bee venom. Phospholipases A2 (PLA2) are enzymes capable of hydrolysing the ester linkage of glycerophospholipids leading to the liberation of free fatty acids and lysophospolipids, including arachidonic acid, a precursor necessary for the biosynthesis of eicosanoids through the intervention of cyclooxygenase, molecules involved in the inflammatory cascade. Phospholipase A2 is therefore involved in the synthesis of prostaglandins and leukotrienes. Pure phospholipase A2 is not poisonous, but when it is near melittin, it becomes a haemolytic factor. [21]. It works in concert with melittin to lyse erythrocytes by causing breaches in the cell membrane that let melittin flow through. Melittin does this by dissolving the phospholipid layers that make up the majority of the cell membranes. This haemolytic effect of bee venom is inhibited by heparin [58]. Additionally, it has been demonstrated that phospholipase A2 has anti-inflammatory, anti-tumor, and anti-parasitic properties [59,60].

### 3.6. Hyaluronidase

Hyaluronidase is a 350 amino acid residue polypeptide that makes up 1% to 2% of the BV. Human hyaluronidase, which is involved in the turnover of hyaluronic acid, and bee venom hyaluronidase share a 30% sequence identity [61]. Hyaluronidase acts as an adjuvant to venom diffusion. Certain acidic mucopolysaccharides in connective tissues have intrinsic glycolide linkages that the enzyme hyaluronidase breaks down, decreasing the viscosity of the tissue and allowing the venom to enter the tissues [62]. Additionally, because hydrolyzed hyaluronic acid particles have pro-inflammatory, pro-angiogenic, and immune-stimulating capabilities, they accelerate systemic poisoning [21]. It is known to promote blood vessel dilatation, increasing permeability and hence, blood circulation, which in turn enhances BV circulation [60].

The Table 2 below summarizes the key characteristics of the bee venom’s constituent parts and the associated biological effects.

## 4. Collection of Bee Venom

The final product has varying properties depending on the extraction process used. The most effective venom appears to be that which is collected under water to prevent the evaporation of the highly volatile components. Venom from venom sacs that had been surgically excised had a different protein composition than venom obtained using the electric shock technique [15].

The venom gland was traditionally removed surgically from bees, or bees were individually squeezed until a drop of venom was extracted from the stinger tip. Electroshock extraction has gradually gained popularity since the early 1960s and is now considered the norm. The bees are induced to release venom by touching wires covered in a fine wire mesh through which low-voltage electric current discharges (20–30 volts) pass; the venom is then periodically collected from a glass plate beneath the wires, such as every ten minutes. Ten thousand stings produce one gram of venom in an hour or two [27].

The method of using electrical shocks with bees to extract their venom was first described by Markovic and Mollnar (1954) [63]. There are different models and installations [64,65,66]. Depending on the author describing the technique, different parameters are sought to collect bee venom. Typically, the voltage is between 24 and 30 volts, the pulse lasts between 2 and 3 s, there is a 3 to 6 s gap, and the pulse frequency is between 50 and 1000 hertz. The bees are not harmed throughout the operation, making it safe for them. One hundred fifty milligrams of honeybee venom can be collected after 3 h of harvesting. Twenty hives may yield one gram of venom over the course of two hours [67]. Four grams of dried bee venom can be acquired if venom is collected three to four times each month for three hours between April and October [68]. However, the price to pay is a 10% to 15% decrease in brood activity and honey production. Bee productivity is unaffected by less frequent harvesting (3 to 4 times each season).

Individual bee venom active ingredients can be extracted for specialized medical and biological purposes using chromatographic separation techniques or molecular genetic techniques [27,69].

Electric shock extraction techniques momentarily disturb hive life because the bees become hostile and produce the alarm pheromone [18,70]. The venom obtained in this manner, known as “apitoxin,” becomes more unstable and loses some of its volatile components when compared to the venom preserved pure in the bee’s sac (the esters, whose therapeutic value is antispasmodic, calming, tonic, anti-arhythmic). When the venom is ready, it can be administered using acupuncture needles that inject the poison right into the affected area. A conventional and affordable method of delivering the active ingredients in their most complete and fresh form involves the bee directly stinging the patient.

## 5. Effects and Applications in Veterinary Medicine

One of the products that has undergone the most extensive investigation in the fields of biology and medicine is bee venom. It is also a traditional medicinal product that is most well-known around the world [71]. Many research investigations have been conducted during the past 20 years employing animal models, particularly murine models. Bee venom or its components are used in apitherapy, allergology, as well as in experimental biology [72,73]. Bee venom has a variety of diverse and perhaps contradictory biological effects. It is necessary to employ certain bee venom components to produce particular biological effects. Like many medications, bee venom has adverse effects that must be taken into account in addition to the intended therapeutic effects. Crude bee venom has substantially lesser toxicity when compared to the combined action of its constituent parts. Bee venom has toxic effects if the dose provided is 200–500 times more than the therapeutic dose, whereas individual bee venom components have toxic effects if the dose used is 20–50 times greater than recommended [74]. Bee venom therapies are currently accessible throughout the world, although they are most popular in Asia, Eastern Europe, and South America. In addition to treating “traditional” musculoskeletal illnesses like arthritis and rheumatism, diverse therapeutic uses also extend into grey areas, as in the case of a new study that suggests bee venom as a supplemental treatment for COVID-19 [75,76,77]. In the next section, the antioxidant, anti-inflammatory, anti-pathogenic, and anti-carcinogenic activities aimed at resolving neurological disorders of bee venom will be analysed (Figure 1).

### 5.1. Antioxidant Activity

Bee venom contains substances with strong antioxidant activity (AOA) [78]. Phospholipase A2, apamin, and melittin are responsible for this reaction. A variety of methods, including free-radical scavenging, hydrogen donation, metal ion chelation, single oxygen quenching, and acting as a substrate for superoxide and hydroxyl radicals, underpin the antioxidant activity. The antioxidant effect may result from the compounds’ ability to inhibit lipid peroxidation (a process involving those chemical species with an independent existence and one or more unpaired electrons or an odd number of electrons, the so-called free radicals) and boost superoxide dismutase activity (an important enzyme that reduces radical damage by removing the superoxide radical in almost all cells exposed to oxygen). However, bee venom also contains additional substances that function as antioxidants in addition to these. For instance, vitellogenin demonstrates antioxidant action in mammalian cells by directly protecting the cell from oxidative stress, providing the cells with a defense against reactive oxygen molecules.

Only three recent investigations have used conventional tests to measure the AOA of bee venom [79,80,81]. Antioxidant qualities were present in all samples, which appeared to be unrelated to any of the specific components that were discovered and measured in the same samples. According to certain evidence, melittin alone exerts very low AOA compared to bee venom extracts, and this may be because other venom components also play a role [82]. Therefore, some other small compounds may be implicated in the observed bioactivities, along with synergistic or antagonistic actions at certain dosages, leading to various outcomes across bee venom samples.

Among the first studies conducted, we have that of Rekka et al. (1990) [83]. Rekka et al. demonstrated that honeybee venom had a considerable inhibitory effect on nonenzymatic lipid peroxidation. Additionally, it has strong hydroxyl radical scavenging abilities, as demonstrated by its competition with dimethyl sulfoxide for HO (hydroxyl radicals). These findings may further support the idea that antioxidant activity plays a role in the anti-inflammatory properties of honey bee venom, which is mostly known for its ability to decrease interleukin-1 production in vitro [83].

Other studies have investigated antioxidant activity in conjunction with other parameters. For example, El-Hanoun et al. (2020) [84] administered 0.1, 0.2, and 0.3 mg per rabbit subcutaneously twice a week for a total of 20 weeks. The total antioxidant capacity (TAC), glutathione S-transferase (GST), glutathione content (GSH), glutathione peroxidase (GPx), superoxide dismutase (SOD), malondialdehyde (MDA), and TBARS were all tested during the experiment to determine any potential changes in the antioxidant activity. The GST and GSH levels in the treated rabbits increased, according to the findings. MDA and TBARS values were also lower. These findings supported BV’s antioxidant properties. Additionally, BV resulted in an enhancement in reproductive performance that was directly associated with the semen’s increased antioxidant activity. Similar evidence of improvement of several parameters was obtained by subcutaneous administration of bee venom in rabbits by Elkomy et al. (2021) [85]. Particularly, compared to the control group, milk yield, litter size at birth, litter weight, and survival rate at weaning age were all considerably higher in the BV groups. When compared to the control group, serum estradiol 17- (E2) levels in the rabbit treated with BV were 15% higher, which was statistically significant. In comparison to the control, the does who were given any study dosages of BV saw a gradual and significant drop (12%) in serum progesterone levels (P4). Additionally, compared to the control group, they demonstrated a statistically significant rise in conception (17%) and fertility rates (10%). When BV was administered to the rabbit, the activity of the liver enzymes aspartate aminotransferase (AST) (16%) and alanine aminotransferase (37%) gradually decreased and became significantly lower. Results also showed that BV caused a significant decrease in malondialdehyde (MDA) and thiobarbituric acid reactive substances (TBARS) in BV groups compared to the control group, as well as a gradual and significant increase in total antioxidant capacity (TAC), antioxidative enzymes like glutathione S-transferase (GST) and glutathione peroxidase (GPx), serum IgG, IgM, and Ig [85].

Instead, the study by El-Speiy et al. (2022) investigated the possibilities of supplementing bee venom (BV) to the drinking water of rabbits [86]. The association of BV with oxytetracycline (OXY) was proposed in an experimental group of investigation. Subsequently, it was looked at how weaned rabbits’ immunological status, bacterial count, antioxidant activity, and haemato-biochemical profile had changed. The findings showed that, except for the rise in ALT in the OXY group compared to the control group, weaned rabbits treated with OXY and BV had significantly higher levels of total plasma protein (TP) and globulin (Glo) while having lower levels of AST and ALT. Triglycerides (TG), total cholesterol (TC), and very low-density lipoprotein (VLDL) were dramatically reduced in rabbits treated with OXY or BV (VLDL-c). IgG, total antioxidant capacity (TAC), superoxide dismutase (SOD), catalase (CAT), and glutathione peroxidase all increased in the BV-treated groups (GPX).

The broiler has also been used to assess the antioxidant capabilities of bee venom. In particular, in the research of Kim et al. (2019) [87], increasing concentrations from 10 to 500 micrograms per kilogram have been added to the basic diet. Numerous serum markers and antioxidant activity were assessed after 21 days. Additionally, the liver was removed to analyze the fatty acid content and quantify the malondialdehyde levels. Except for triglycerides and non-esterified fatty acids, the blood parameters did not change by adding dietary bee venom. Bee venom consumption quadratically enhanced the content of stearic acid in the diet but also lowered the concentrations of palmitoleic, oleic, and linoleic acids as well as monounsaturated and polyunsaturated fatty acids. Finally, eating bee venom tended to quadratically reduce hepatic malondialdehyde levels. The authors concluded that eating bee venom enhanced antioxidant capacity and changed the metabolism of fatty acids in broilers [87].

### 5.2. Antimicrobial and Antiviral Activity

Antimicrobial resistance is a global public health challenge, accelerated by antibiotic misuse [88]. Infections caused by multi-drug resistance bacteria have resulted in an increase in fatalities in recent years, and the issue seems to be getting worse [89]. These resistant bacteria are also a concern in the food chain, as the bacteria can resist common biocides used in the food industry and reach consumers [90]. In such a scenario, bee venom proves to be a promising ally. The peptide melittin is primarily responsible for the antibacterial properties of bee venom [91]. Melittin’s capacity to disrupt cellular membranes serves as the primary mechanism of its antibacterial effect. Due to the structure of the cell membrane, Gram-positive bacteria are more sensitive to melittin than Gram-negative bacteria. Melittin can more readily pass through the peptidoglycan layer of Gram-positive cells’ membrane than it can through the lipopolysaccharide layer that protects Gram-negative cells’ membrane. According to Fennell et al. (1968) [45], melittin and bee venom were effective at killing 86% of Gram-positive bacteria and 46% of Gram-negative bacteria. One milligram of melittin has the same antibacterial action as 0.1 to 93 units of penicillin on Gram-positive microorganisms. It has been demonstrated that the proline residue at position 14 is crucial to melittin’s antimicrobial action. Its absence in a melittin analogue significantly reduced the anti-microbial activity compared to the native peptide.

Melittin’s effectiveness as an antibacterial agent has been investigated against several bacteria, including *Escherichia coli*, *S. aureus*, and *B. burgdorferi* [92,93,94,95]. At MICs of 0.5–4, 0.5–4, and 1–8 g/mL, respectively, melittin shows antibacterial activity against methicillin-sensitive *S. aureus* (MSSA), MRSA, and *Enterococcus spp*. bacteria [96]. *B. burgdorferi*, the cause of Lyme disease, has been found to be affected by both BV and melittin alone in terms of the form and size of its biofilms [93]. BV’s antibacterial and antibiofilm properties against 16 *Salmonella* strains obtained from chickens were also investigated. The MIC for BV was between 256 and 1024 g/mL. In 14 of the 16 *Salmonella* strains that were examined, sub-inhibitory doses of BV considerably decreased biofilm production while significantly increasing motility. The motility of *Salmonella isangi IG1* and *S. infantis Lhica I17* was unaffected by BV [97]. The therapeutic efficacy of bee venom (BV) was also investigated against clinical and subclinical mastitis in dairy cows. In the study by Han et al. (2009), mastitic cows selected on the basis of a somatic cell count above 200,000 cells/mL milk were used [98]. Fifteen lactating mastitic cows were subcutaneously injected with four different BV dosages (3, 6, 12, and 24 mg per treatment) to examine the effects of the BV dose. It was shown that the action of BV enhanced the number of clinically cured quarters with less than 0.2 million/mL somatic cell count (SCC) during the duration of the treatment period. Within two weeks of receiving BV therapy, a considerable decrease in the detection of *Staphylococcus aureus* and other Gram-positive bacteria was seen. The authors concluded that the mammary defense systems of dairy cows with mastitis may have been compromised by BV therapy [98]. The findings demonstrated that bee venom significantly inhibits seven main bacterial mastitis pathogens. In comparison to normal pharmaceuticals, the minimal inhibitory concentrations (MIC) against methicillin-resistant *Staphylococcus aureus* (MRSA), *Staphylococcus aureus*, and *Escherichia coli* exhibited a greater effect. Therefore, apitoxin may have antibacterial properties, which justifies testing it as an alternative to antibiotics for the treatment of bovine mastitis.

Bee venom has proven to be equally effective in the treatment of canine bacterial otitis. In particular, it was observed that dogs treated with apitoxin injection three times a week for a fortnight had a similar bacterial count to the experimental control group treated with conventional antibiotics [99].

Along with other similarly active components, the enzyme phospholipase A2 also possesses antibacterial characteristics [100]. In addition, vitellogenin acts as an antimicrobial peptide, causing damage to the cell membranes of bacteria [71]. The enzyme phospholipase A2 also has antifungal properties against some species of the *Candida* genus and can also be employed as an anti-parasitic drug in the treatment of specific organisms like *Trypanosoma* and *Plasmodium falciparum* [101,102,103]. Studies on the use of bee venom against *Toxoplasma gondii* should also be highlighted. Bee venom exerts damaging effects on live tachyzoites, as shown by Hegazi et al. (2014) [104].

It is important to concentrate our attention on viruses, which are likewise deserving of discussion. Studies on animal and plant viruses have already shown mellitin’s capacity to damage cell membranes and interact with cell surface molecules; the latter is a key component in antiviral therapy [43]. Mellitin has proven to be effective against Papillomaviruses and the vesicular stomatitis virus [105,106]. Mellitin has also shown antiviral properties against viruses without a viral membrane. Through a significant up-regulation of Th1 cytokines (IFN- and IL-12) and several immune cell types, including CD3+ CD8+ T-cells, CD4+CD8+, BV and its component melittin, can induce immunity against porcine reproductive and respiratory syndrome viruses (PRRSV), resulting in a decrease in viral load and a milder form of interstitial pneumonia in pigs infected with PRRSV [107].

A final consideration can be made about the use of antibiotics for auxinic purposes, aimed at preventing diseases and promoting weight gain in animals with low dosage administrations. This use of antibiotics was banned in the European Union in 2006, but was used in the past [108]. The study conducted by Kim et al. (2018) was designed to evaluate purified bee venom (BV) as an alternative to antibiotics in broilers [109]. To obtain 0, 10, 50, 100, and 500 g of BV per kg of food, BV was added to a diet of soybean meal and corn meal. Dietary BV enhanced body weight gain to 1–21 days as the amount in the diet improved the feed conversion ratio quadratically. Secretory immunoglobulin A (sIgA) content on the ileal mucosa rose linearly with the addition of BV to the diet at 21 days and 1.5 months. With increasing toxicity in the meals at 21 days, the total amount of short-chain fatty acids (SCFA) in the cecal digesta decreased. Except for creatinine, none of the serum values were impacted by the BV diet [109]. Similarly, Han et al. (2010), who added bee venom to water acquired comparable proof of weight growth in broilers [110]. Finally, positive effects on growth performance and immunocompetence blood parameters were also recorded in young pigs [111]. Bee venom could also be interesting in this capacity as a growth promoter.

### 5.3. Anti-Inflammatory Activity

Bee venom contains at least four substances that have anti-inflammatory effects. Among the most important, we have melittin, apamin, adolapin, and phospholipase A2. Mellitin has been extensively studied for its anti-inflammatory properties against liver inflammation, amyotrophic lateral sclerosis, atherosclerosis, and neuroinflammation [112,113,114,115].

Inflammation can largely be seen as the manifestation of a genetic transcriptional programme activated by tissue danger signals. Nuclear factor-kappa B (NF-κB) is a protein complex functioning as a transcription factor. NF-kB plays a significant role in the regulation of inflammatory genes (cyclooxygenase-2 (COX-2), cytosolic phosphatase A2 (cPLA2), and tumor necrotic factor-a) [116]. It is located in the cytosol, bound to an inhibitory protein called inhibitor of κB (IκB). A variety of extracellular signals reaching the membrane receptors of the Toll family can activate the enzyme IκB kinase (IKK). IκB kinase (IKK) phosphorylates IκB, which detaches. NF-κB is then free to migrate to the nucleus, where it binds to NF-κB consensus sites located upstream of several inflammatory genes. The inhibition of nuclear factor kappa B (NF-B) by the enzyme kinase IkB (IKK) may be how the possible anti-inflammatory mechanism function works. IKK activity is inhibited by melittin’s interaction with the enzyme. These protein–protein interactions were linked to the inhibition of IKKa and IKKb activity as well as the prevention of IkB release in response to an inflammatory stimulation [116]. Inhibition of IKK reduces the production of phospho-p38, the cytokines interleukin-1 (IL-1), and interleukin-6 (IL-6), as well as reducing the release of tumour necrosis factor (TNF) [113,117,118].

Melittin significantly inhibits MAPKs (Mitogen-activated protein kinases), including ERK and p38 MAPK [119]. MAPKs (Mitogen-activated protein kinases) are enzymes widely present in the body and involved in numerous physiological and pathological processes. In particular, p38 represents a very interesting pharmacological target, being a protein involved in various cellular responses. Its enzymatic activity occurs both in the cytoplasm and in the nucleus. It is involved in various cellular processes, such as the regulation of mRNA, apoptosis, protein degradation, and the organisation of the cytoskeleton and chromatin. The activation of p38, in particular p38α, has been correlated with inflammation by in vivo and in vitro studies, which have demonstrated its ability to intervene at transcriptional and post-transcriptional levels. It induces COX-2 activity and regulates the biosynthesis of numerous proinflammatory cytokines (IL-1β, IL-6, TNF-α, INF-gamma) that are fundamental in the pathogenesis of autoimmune diseases (such as rheumatoid arthritis, multiple sclerosis, Crohn’s disease), asthma and COPD (chronic obstructive pulmonary disease), but also of cardiovascular disorders such as atherosclerosis [120,121,122].

Therefore, treatments with melittin modulate TLR pathway activation and prevent inflammatory cytokine expression [123]. Melittin has the ability to block nuclear NF-kB p65 activation and MAPK serine p38 inhibition in vitro [124]. As a result, this function has anti-inflammatory effects. By modifying the transcription factors NF-kB and AP-1 in vivo, melittin also showed anti-inflammatory capabilities [125]. Recent research has revealed that using BV topically for atopic dermatitis has anti-inflammatory effects. IgE concentrations, cytokine release, and the activity of NF-kB and MAP kinases are all decreased, which has an anti-inflammatory impact. Both the TNF-/IFN-dependent inflammatory response and the lipopolysaccharide (LPS)-induced inflammatory responses are inhibited by the lowering of NF-kB and MAPK activity. A decrease in MAPK activity also affects the expression of the inflammatory genes COX-2 and iNOS, as well as the control of NF-kB signals that affect cytokine release [126,127]. Studies on the anti-inflammatory properties are among the most numerous. A large portion of the controlled clinical research on how bee venom affects arthritis has been carried out on arthritic animals (mice, rats, and guinea pigs). One of the most popular test models in scientific research to create new substances that can be utilized to treat arthritis and other inflammatory disorders is adjuvant-induced arthritic rats.

Bee venom was used in investigations on the prevention and treatment of adjuvant-induced arthritis in rats by Lorenzetti et al. (1972) [128]. The authors demonstrated that administering bee venom subcutaneously to arthritic rats three times a week has a significant impact on both avoiding the onset of arthritis and reducing the severity of an already present case. The anti-inflammatory impact of bee venom, according to these investigators, was far more prominent when it was applied as a preventative measure.

Similar findings have emerged in subsequent studies. According to Zurier et al. (1973) [129], daily injections of three distinct bee venom components—melittin, apamin, and phospholipase A 2—along with daily injections of entire bee venom, delayed the onset of adjuvant arthritis in rats.

Chang and Bliven (1979) [130] investigated the mechanism of action of bee venom on adjuvant-induced arthritis and came to the conclusion that at least two processes were involved. They discovered that a single bee venom injection given the day before or the day of the adjuvant injection significantly prevented the development of arthritis in rats. According to their findings, this suppression was caused by an altered immune response in addition to the bee venom’s anti-inflammatory effects [130]. In the following years, numerous independent studies on the anti-inflammatory benefits of entire bee venom have been conducted using rat models. In the rat adjuvant-induced arthritis and carrageenan footpad edema tests, these investigations showed that entire bee venom has an anti-inflammatory effect. Chang and Bliven (1979) [130] discovered that using these in vivo assays that bee venom (0.5 mg/rat subcutaneously) and cyclophosphamide (60 mg/kg orally) behaved similarly in suppressing adjuvant-induced arthritis in rats and showed a distinctly different temporal pattern of activity from steroid therapy, even when considering the theory that the effects were caused by stress-induced steroid production. Additionally, their findings indicated that this suppression may have been caused by changes in the immune system, in addition to the bee venom’s anti-inflammatory effects.

Eiseman et al. (1982) [131] investigated how bee venom affected the progression of adjuvant-induced arthritis and drug metabolism depression in rats. The main and secondary inflammatory responses to the adjuvant hind paws were shown to be reduced by bee venom, according to the investigators. Additionally, they noticed variations in heme metabolism brought on by the venom, which is strongly suggestive of immune system disturbances leading to modifications in the hepatic microsomal enzymes.

Studies in mouse models were followed by clinical trials in animals. Melittin contains anti-inflammatory effects that activate the pituitary-adrenal system and causes it to release cortisol, as Vick et al. showed in monkeys as early as 1972 [132]. Based on findings from animal models, it is estimated to be 100 times more effective than hydrocortisone [132].

Another study by Vick et al. (1976) describes an application of bee venom in the treatment of canine arthritis [133]. In particular, a group of 24 dogs was enlisted, of whom 8 were verified to have arthritis by radiographic examination and a thorough physical examination, and 16 were randomly chosen as normal dogs. On days 30, 37, 50, and 60, 1 mg of whole bee venom was given subcutaneously to the arthritic dogs. Weekly assessments of motor activity in the cage were made using a pedometer, and plasma cortisol levels was registered. Plasma cortisol levels rose 15 days following bee venom injection treatment. Prior to therapy, the arthritic dogs travelled, on average, about four miles per day compared to the normal canines, who covered roughly twelve miles per day. The average activity of the arthritic dogs resembled that of the healthy dog population after just three doses of bee venom. Additionally, this increase in activity persisted for sixty to ninety days after the final injection, suggesting that the venom had a long-lasting impact [133].

Results of the treatment of 17 arthritic dogs were reported by Short et al. in 1979 [134]. Fourteen out of seventeen dogs considerably recovered after receiving bee venom treatment, recovering to normal or almost normal mobility. All dogs with disc complications recovered to normal or nearly normal conditions after a series of bee venom injections were given at the sites of pain and stiffness. Four of five dogs treated for joint complications (hip dysplasia and arthritic joints) showed improved movement. Four of six dogs treated for poor surgical recovery responded well. The authors of this study came to the conclusion that some canine arthritic disorders may be greatly improved by bee venom treatment.

Anti-inflammatory effects have also been studied in large animals. In the study of Jeong (2017), apitoxin was employed in mastitis treatment [135]. The idea was that BV would lessen inflammation in the cells of the cow mammary epithelium (MAC-T). Cells were treated with LPS (1 g/mL) to cause an inflammatory response, and the anti-inflammatory effects of BV (2.5 and 5 g/mL) were investigated to test the theory. The cellular defenses of BV against LPS-induced inflammation were also investigated. The findings demonstrated that BV can reduce the production of COX2, a protein associated with inflammation, as well as pro-inflammatory cytokines, including IL-6 and TNF-. In LPS-treated cells, BV dramatically reduced the activation of NF-B, an inflammatory transcription factor, via dephosphorylating ERK1/2. Additionally, pretreatment with BV reduced the amount of intracellular reactive oxygen species that were LPS-induced (e.g., superoxide anion). These results support the idea that BV may suppress oxidative stress, NF-B, ERK1/2, and COX-2 signaling to diminish LPS-induced inflammatory responses in bovine mammary epithelial cells [135].

In research by Von Bredow et al. (1978) [136], the authors evaluated the effects of bee venom injections on eight arthritic horses with ages ranging from 8 to 17. Three of the six horses who showed a significant improvement—out of the eight, six—showed full recovery. Finally, bee venom has also been investigated against important horse diseases. Kim et al. (2006) [137] describe a case of a 13-year-old male Arabian horse suffering from laminitis. The subject was injected with bee venom at several sites. After the third session, the patient showed almost normal walking.

Between 12 and 25% of the racing population of horses were estimated to have chronic obstructive pulmonary disease (COPD). However, since the fibrobronchoscope was developed, it is now conservatively believed that 50% of horses at racetracks have COPD. This ailment needs to be taken into account if a horse performs below standard. Since drugs are prohibited in racing, COPD poses a significant concern for racehorses. The effectiveness of bee venom in treating COPD in racehorses was the subject of a randomised controlled trial. From among the horses competing on the Ontario Jockey Club circuit, some were chosen at random. In total, 3.0 milliliters of bee venom were subcutaneously injected, 0.5 milliliters per site, into three bilateral acupuncture points. The performance of 76% of the horses improved as their COPD decreased [138]. Therefore, bee venom has also demonstrated valuable properties in this direction.

### 5.4. Neurodegenerative Disorders

The neuroinflammation caused by persistent glial cell and microglial activation is associated with neurodegenerative diseases. Pla2 and apamin, two components of BV, have been investigated as anti-neuroinflammatory medicines, and as an adjuvant to increase the effectiveness of some medications against neurodegenerative illnesses and/or to lessen adverse effects [139]. The use of BV was investigated in neurodegenerative diseases by Tsai et al. in 2015 [140]. To treat dogs with intervertebral disc disease, the authors used bee venom. Intervertebral disc disease is a disorder characterized by algia that calls for the use of analgesics and anti-inflammatory medications to lessen pain and nociceptive signals. The study’s authors found that BVA efficiently reduces pain symptoms. Forty dogs with neurological conditions brought on by intervertebral disc disease were enlisted and split into experimental groups to get this information. The myelopathy scoring system and the functional number scale scores of the dogs receiving BV treatment dramatically improved and were comparable to those of the group receiving traditional therapy [140].

Other research has focused on non-immune mediated nerve palsy. For example, Jun et al. (2007) [141] studied the effects of bee venom on facial nerve paralysis (FNP) [141]. Twelve dogs were enlisted for this study and split into three experimental groups: a control group with four dogs, a dexamethasone group with four dogs, and a bee venom group with four dogs. In the control group, saline solution (1 mL) was intramuscularly injected into the head muscle following FNP induction. Clinical ratings, drawn up in the form of treatment points, were used to track changes in the clinical symptoms of FNP after the injection of 100 g apitoxin. Bee venom was successful in resolving the condition. Comparisons were also made between the groups treated with apitoxin and dexamethasone. However, there was no discernible difference between the groups treated with apitoxin and dexamethasone [141]. The same outcome was attained in an apitoxin-treated 6-year-old male Shih Tzu dog that had the left side of his face paralyzed and tilted. Following bee venom acupuncture therapy, the clinical symptoms gradually became better. Eight weeks after beginning bee venom acupuncture, the patient’s face sensory and neurological indicators improved [142]. This case study demonstrates the potential benefit of using bee venom in acupuncture for canine idiopathic facial paralysis.

It has also been demonstrated that BV is effective in treating peripheral neuropathy that develops in association with vincristine therapy. Due to damage to the sensory and motor neurons in the peripheral nervous system, several chemotherapy drugs do cause peripheral neuropathy. The effects and mechanism of bee venom injection (BVA) were examined in the study by Li et al. (2020) to treat peripheral neuropathic pain brought on by repeated intraperitoneal vincristine (1 mg/kg/day, days 1–5 and 8–12) infusions in rats [143]. Bee venom (BV, 1.0 mg/kg) administered subcutaneously at therapeutic site ST36 reduced mechanical and cold sensitivities. Extracellular vivo recording demonstrated that bee venom acupuncture (BVA) prevented vincristine-treated rats’ abnormal spinal wide-range dynamic (WDR) neuron hyperexcitation brought on by cold cutaneous (acetone) and mechanical (brush, push, and pinch) stimuli. Additionally, BVA’s effects on mechanical and cold hypersensitivity were reversed by lidocaine microinjection into the ipsilateral locus coeruleus or by antagonists of spinal 2-adrenergic receptors, demonstrating the critical role of descending noradrenergic regulation in analgesia [143]. These results suggest that BVA could be a potential therapeutic option for vincristine-induced peripheral neuropathy.

### 5.5. Anti-Cancer Activity

In recent years, there has been a great deal of interest in finding natural compounds having anti-cancer potential. Mellitin and phospholipase A2 are the components of bee venom that have antitumor action [144,145,146]. The interactions between them also lead to other antitumor effects, such as the production of apoptosis and necrosis, as well as the suppression of the proliferation of different tumor cells.

The most appealing action for slowing the proliferation of cancer cells is apoptosis. The part of BV with the highest level of cytotoxic action against cancer cells is melittin. The first study establishing the antitumor impact of melittin revealed that leukemic cells’ inhibition of calmodulin caused cancer cells to undergo apoptosis. This Ca^2+^ channel pump blockage led to a massive increase in Ca^2+^ concentration, which ultimately caused cell death [147]. Since that time, several investigations on the antitumor effects of melittin and their mechanisms of action have been carried out using various types of tumor cell lines.

Nowadays, it is known that different bee venom components stimulate various and distinct cell signaling pathways in different ways. At the cell surface, many growth factor receptors (epidermal growth factor receptor, TNF receptor, etc.) are activated during carcinogenesis. These receptors’ activation triggers multiple downstream signaling cascades. The Ras-MAPK (including ERK and JNK) pathways, the PI3K-AKT pathways, PLC-γ-CaM, and the NF-kB are significant and are the targets of bee venom components among these pathways. Some components of bee venom inhibit surface receptors either by dephosphorylating them or by causing their degradation, which in turn modifies the signaling pathways downstream that are crucial for proliferation, metastasis, angiogenesis, and apoptosis (for example, the synergistic effect of BV sPLA2 and PtdIns(3,4)P2); PtdIns(3,4)P2 and bv-sPLA2 are responsible for the cytotoxic action because they cause cell death as a result of membrane integrity loss, absence of signal transmission, and production of cytotoxic lyso-PtdIns(3,4)P2. Bee venom components frequently inhibit AKT and ERK signaling, albeit frequently, this inhibition is the consequence of growth factor inhibition. Another typical target of bee venom components is the inhibition of the NF-kB signaling pathway by interfering with several signaling targets. Certain bee venom components produce reactive oxygen species (ROS), which activate members of the p53 family and cause cell cycle arrest and death [148]. The anti-cancer activity of bee venom is also expressed in its ability to modulate cell apoptosis.

The growth of multicellular organisms and the maintenance of good tissue homeostasis both depend on apoptosis. Two molecular programs that ultimately result in the activation of particular members of the caspase family are in charge of controlling the execution of this route in mammals. This subsequently results in the cleavage of key cell substrates, leading to cell death. The intrinsic pathway, which is triggered by a wide variety of stress signals, and the extrinsic pathway, which functions downstream of death receptors like Fas and the tumour necrosis factor receptor family, are the two molecular programs. The identification of cytochrome c as an apoptogenic factor released by mitochondria marked a crucial breakthrough in the discovery of the importance of these organelles in the intrinsic pathway of apoptosis. The ‘point of no return’ in this pathway is defined by mitochondrial outer membrane permeabilisation (MOMP), which leads to the release of cytochrome c.

The intrinsic (mitochondrial) pathway of apoptosis is controlled by the BCL-2 family of proteins, which operate within a complex network of protein-protein and protein-membrane interactions. The BCL-2 family includes both pro- and anti-apoptotic regulators of the intrinsic apoptosis pathway. The BCL-2 family of proteins regulates MOMP and thus determines the cellular commitment to apoptosis. Indeed, BCL-2 proteins play a key role in mediating the delicate balance between cell survival and death.

Several cytotoxic stimuli, including oncogenic stress and chemotherapeutic agents, as well as developmental signals, involve the mitochondrial pathway, which is regulated by members of the BCL-2 family. These stimuli activate BH3-only (initiator) proteins, which inhibit pro-apoptotic BCL-2 proteins (guardians), thus allowing the activation of the pro-apoptotic effectors BAX and BAK, which subsequently disrupt the outer mitochondrial membrane. Cytochrome c released from the mitochondria promotes caspase-9 activation on the scaffold protein Apaf-1, while the released protein SMAC (second mitochondria-derived activator of caspases) blocks the caspase inhibitor XIAP (X-linked inhibitor of apoptosis protein). BH3-only proteins, which are transcriptionally or post-translationally induced by cytotoxic stress signals, exert their pro-apoptotic function by two mechanisms: the neutralisation of pro-apoptotic BCL-2 proteins and the direct activation of the pro-apoptotic effectors BAX and BAK. Bee venom and melittin have been shown to increase the expression and levels of a number of pro- and apoptotic mediators, including cytochrome C (Cyt C), protein 53 (p53), Bcl-2-associated X protein (Bax), Bcl-2 homologues antagonist/killer (Bak), caspase-3, caspase-9, and various death receptors, while decreasing the anti-apoptotic mediator Bcl-2 [17,148,149]. Other mechanisms are involved in the apoptotic activity of bee venom. The expression of metalloproteinases, enzymes involved in the breakdown of collagen and thus in pathological invasion processes (tumour metastasis), is suppressed by bee venom. Matrix metalloproteinase-9 (MMP-9) expression and activity can be suppressed by inhibiting extracellular/mitogen-activated protein kinase p38 (ERK/p38 MAP) regulated protein kinases and NF-B pathways [150,151]. 

Furthermore, in relation to tumour progression, by suppressing vascular endothelial growth factor (VEGF), VEGFR-2, and preventing the latter’s signaling pathways, bee venom significantly reduced angiogenesis and metastasis [152]. Melittin has been shown to increase [Ca^2+^] and activate phospholipase A2 (PLA2) in numerous cancer cell types. Melittin’s membrane-disrupting activity causes cell membranes to become more permeable, which appears to open specific kinds of Ca^2+^ channels and raise intracellular Ca^2+^ concentration. Apoptosis or necrosis of mammalian cells is then brought on by changes in the cytosolic free Ca^2+^ level [148]. In prostate cancer, BV and melittin demonstrated tumor cell growth inhibition. The down-regulation of anti-apoptotic gene products such as Bcl-2, XIAP, iNOS, and COX-2 was the source of this impact [17,153]. As a result, this down-regulation inhibited NF-kB transcriptional activity, which is correlated with apoptosis, the process through which cells die. By preventing the translocation of p50 and p65, IkB phosphorylation was impaired, which in turn led to the deactivation of NF-kB signalling [17]. In ovarian cancer cells, a different mechanism of BV and melittin-induced apoptosis was investigated. Melittin had an antitumor impact in this study by activating death receptors and inhibiting the JAK2/STAT3 pathway [154]. Inactivation of STAT3 and increased expression of the death receptors DR3, DR4, and DR6 were the major strategies used to slow the proliferation of ovarian cancer cells. When these DRs were expressed, apoptosis was activated in a caspase-8-dependent manner [155].

Alternately, melittin employs additional cancer-fighting pathways in addition to apoptosis. Melittin reduced tumor cell growth without apoptosis in murine animal models of Lewis lung carcinoma. The amount of tumor-associated macrophages (TAMs), particularly CD206+ M2-like TAMs in the tumor stroma, was decreased by melittin therapy. Because there were less CD206 + M2-like TAMs, there were fewer VEGF+ and CD31+ cells in the tumor tissue. This fact reveals melittin’s anti-angiogenic properties [155]. The capacity of melittin to interact with phospholipid membranes is linked to further processes that result in the death of cancer cells. This interaction results in holes that have the potential to cause cell lysis by collapsing the cell membrane [144,156]. This capability was investigated in relation to colorectal and stomach cancer. Melittin produced granulation, blebbing, and cell swelling in cancer cells in vitro in about one minute at a dosage of 20 g/mL, which was very quick. Additionally, full death happened 15 min after the commencement of the therapy [157].

This lytic action, meanwhile, is not limited to cancer cells and can also result in the lysis of healthy cells. In this regard, using vectors like nanoparticles to restrict melittin’s activity in target cells is one possible answer. In a recent study, melittin and nanographene oxide, as well as melittin and nanodiamonds were combined to combat breast cancer [158]. Compared to melittin alone, this combination increased the harmful impact on cancer cells. Furthermore, melittin with nanodiamonds was able to shield healthy cells from melittin’s lytic action. Additionally, it was seen that the degree of necrosis had diminished.

### 5.6. Activities on Other Diseases

Bee venom and melittin have anti-inflammatory effects on digestive disorders linked to inflammation, such as gastric ulceration, nonalcoholic steatohepatitis, ulcerative colitis, acute liver failure, etc.

When acetylsalicylic acid caused gastric ulcers in rats, BV demonstrated gastro-protective properties. In acetylsalicylic acid-treated rats, it attenuated the haematological, haemostatic, and histopathological changes, decreased tissue eosinophil levels, as well as the ulcer index, fluid volumes, and pepsin concentrations. This was attributed to its anti-inflammatory and antioxidant effects by lowering TNF-, MPO levels, and MDA concentration, and increasing SOD activity and glutathione (GSH) concentration, as well as its anti-apoptotic property by downregulating Bax and Caspase-3 level [159].

In streptozotocin-induced diabetic rats, beeswax-coated water-soluble fraction of BV (BWCBVA) reduced blood glucose levels, restored serum biochemical markers, and raised body weight. Additionally, BWCBVA treatment raised the number of islet cells and decreased the damage to cells in the pancreas while decreasing the expression of PI3K-p85 and glucokinase in the liver. Additionally, by boosting calcium ion influx and blocking the potassium ion channel, co-administering BWCBVA with nifedipine and nicorandil caused insulin secretion [160].

The kidney has also been shown to benefit from the effects of bee venom. Acute kidney injury and renal fibrosis have been demonstrated to be protected by BV and its key ingredients, such as MEL and apamin. LPS endotoxin causes systemic inflammatory reactions in sepsis. By reducing oxidative stress, inflammation, and cell death in mice, BV, MEL, and apamin reduced the acute kidney injury brought on by LPS. Consider MEL as an example. It reduced structural and functional damage to the kidneys while attenuating the levels of direct tubular injury indicators in LPS-treated mice. MEL reduced the buildup of immune cells in the kidney and inhibited the NF-B pathway in addition to lowering TNF- and IL-6 levels in the body and the kidney. Following LPS treatment, MEL decreased MDA levels, suppressed the production of nicotinamide adenine dinucleotide phosphate oxidase 4 (NOX4), supported Nrf2-mediated antioxidant defenses, and avoided apoptosis and necrosis [161,162,163].

## 6. Current Limits in Use

Despite its potential for therapeutic benefit, potential side effects and allergic responses connected to its composition must always be taken into account. The development of safe practices depends on this factor. The investigations on the usage of bee venom and its harmful effects were summarized by Park et al. (2015) in a comprehensive literature review [164]. The study’s analysis of 145 articles revealed that 28.87% of patients receiving bee venom had negative side effects. The same evaluation highlighted the subpar nature of research conducted in this field and the subsequent difficulties in result analysis. Additionally, due to the allergenicity of the ingredients, both allergic and anaphylactic reactions happened. Noble and Armstrong (1999) [165] observed clinical signs such as lethargy, haematuria, ataxia, and convulsions in two dogs that had been poisoned by bee stings. One of the two dogs succumbed to death. Nair et al. (2019) [166] observed acute and delayed onset of haemolytic anaemia, echinocytosis, spherocytosis, thrombocytopenia, haemoglobinaemia, and haemoglobinuria following poisoning. Kaplinsky et al. (1977) [167] observed in dogs and cats that high doses of bee venom (above 1 mg per kg) cause an immediate drop in blood pressure to irreversible shock levels. Several types of changes in the electrocardiogram (ECG) were noted: marked shifts of the ST-T segments, the occurrence of varying degrees of atrioventricular block, and severe ventricular arrhythmias. Therefore, a possible contraindication is not knowing whether the patient is allergic to bee venom or not.

In light of the comprehensiveness of the overview provided, the following are some recommended solutions and holes in the area to be filled. First of all, the purification of bee venom is a crucial step. For various applications, it is vital to produce purified bee venom, which is free of histamine and phospholipase A2, to reduce allergic responses and other negative consequences while keeping significant anti-inflammatory efficacy. Secondarily, some components such as melittin have no specific action. In addition to lysing red blood cells, melittin has shown cytotoxicity against both normal and malignant cells. More specialized analogues for therapeutic use can be created by researching melittin’s effects in model systems. The utilization of model systems and live cells can give information to stimulate the invention of novel melittin analogues or hybrid peptides for therapeutic reasons, despite the apparent differences between the actions of melittin in model systems and living cells.

Moreover, it is important to accurately identify and measure the components of bee venom; it is necessary to use processes that allow one to be certain of the substance one is working with. Regarding the latter consideration, standardizing bee venom is necessary to ensure the reproducibility of the outcome. Another aspect to consider is the relatively short plasma half-life of bee venom and the problem of determining the final dose. These critical issues can be overcome with the combination of polymers or nanoparticles [168]. Finally, to increase the effectiveness and safety of the carried substance and lower the likelihood of negative consequences, it is important to understand the precise paths followed by the bee venom or its components follow in the body.

## 7. Conclusions

Bee venom proves to be a promising remedy and/or adjunct for the treatment of numerous ailments. Its antibacterial, antiviral, and anti-parasitic capacities make it extremely attractive for reducing the use of conventional drugs and associated drug-resistance phenomena. Moreover, its use for one condition can also have positive implications for other concomitant diseases, as shown in this in-depth dissertation on its anti-inflammatory, anti-tumour, etc., properties. Certainly, its use in the clinical setting must follow careful efficacy studies aimed at determining the best doses to achieve the pharmacological response while avoiding the side effects associated with its use. As in human medicine, therefore, the already numerous in vitro studies must be followed by clinical trials to determine a conscious and effective use of bee venom in veterinary clinical practice.

## Figures and Tables

**Figure 1 vetsci-10-00119-f001:**
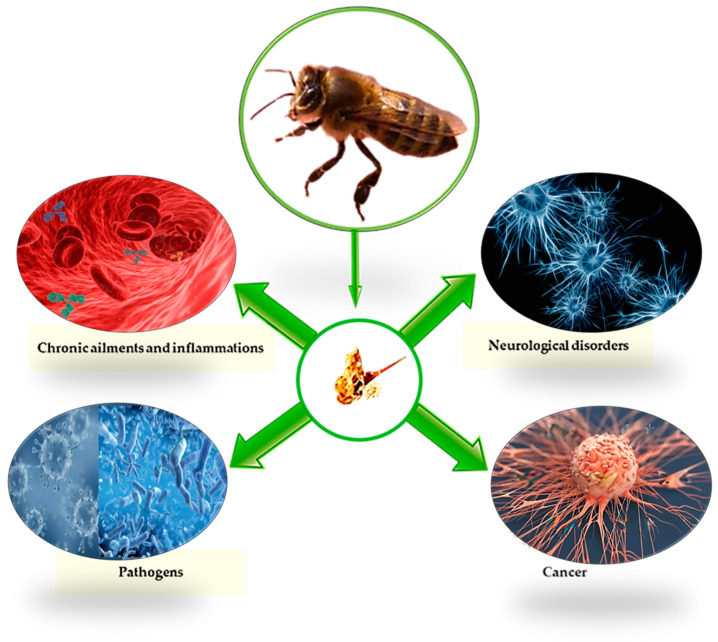
Schematic illustration of bee venom properties.

**Table 1 vetsci-10-00119-t001:** Composition of bee venom: dry matter data according to Dotimas and Hider (1987) [27].

Substance	%	Substance	%
**Enzymes**		**Biogenic amines**	
Phospholipase A2	10–12	Histamine	0.5–2
Hyaluronidase	1.5–2	Dopamine	0.2–1
Phosphatase, glucosidase	1-2	Noradrenaline	0.1–0.5
**Proteins**		**Carbohydrates**	
Mast cell degranulating Peptide	1–2	Sugar (glucose, fructose)	2
Melittin	40–50	**Phospholipids**	5
		**Amino acids**	
Secapine	0.5	Aminobutyric acid and α-amino acids	0.4
Tertiapine, apamin, procamine	2–5	**Volatile substances** (pheromones)	4–8
**Other small peptides**	13–15	**Mineral substances**	3–4

**Table 2 vetsci-10-00119-t002:** Biological effects of bee venom and its components.

Components	Effect
Melittin	Peptide with biological activity. Melittin prevents blood from clotting, works well against germs, shields against radiation. Melittin works as an anti-inflammatory in small dosages. It has a haemolytic action and is clearly cytotoxic.
Phospholipase A2	Phospholipase is the most important allergen and therefore the most harmful component of bee venom.
Hyaluronidase	It is an enzyme that allows venom to enter tissues and causes blood vessels to widen and tissues to become more permeable, increasing blood flow.
Acid phosphatase	Allergen.
Apamin	Biologically active peptide; a neurotoxin.
Mast cell degranulating peptide	Peptide that degranulates mast cells by releasing biogenic amines.
Protease inhibitor	It has anti-inflammatory and hemorrhagic properties and inhibits the action of various proteases, including trypsin, chymotrypsin, plasmin, and thrombin.
Adolapin	Anti-inflammatory, anti-rheumatic, and analgesic.
Histamine	It dilates blood vessels and increases capillary permeability. It is an allergen.
Dopamine, noradrenaline	Neurotransmitters that affect the behaviour and physiology of the senses.
Alarm pheromone	It puts the colony on high alert.

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
