# Peer review of "Therapeutic Use of Bee Venom and Potential Applications in Veterinary Medicine"

_vetsci, 2023, doi:10.3390/vetsci10020119_

Round 1

Reviewer 1 Report

This review provides an overview of the history of bee venom therapy, its production and extraction as well as its application in inflammatory, infectious and cancer processes with a focus on veterinary treatments. Despite numerous reviews on the subject, it is worth publishing this article for its original focus: use of bee venom in veterinary medicine as well as interesting information about production and extraction of bee venom.

I suggest  to the authors to read Khalil et al 2021 (Bee Venom: From Venom to Drug. Molecules 2021, 26, 4941) an interesting and recent review,, which presents an important and recent bibliographic survey of the subject, focusing on pharmacological activities.

References are from peer-reviewed journals, however I am not familiar with specific beekeeping publications.

General comments

Write the names of the bacteria in italics.

Normalize the writing of PLA2 in the text.

Specific comments

Line 90: indicates the references for the diseases described in the paragraph.

Line 157: repetition of sentences in the paragraph.

L 165, Typing error: “carboxy-terminal region”.

L183, Typing error: apamin  is a neurotoxin.

Line 263, indicates the reference of comparison of extraction method.

Line 292, indicates the references of the antioxidant activities of toxins cited in the paragraph.

Line 323, the sentence seems related to the following item 5.2. Antimicrobial and antiviral activity.

Line 332, indicates the reference to melittin's mechanism of action.

Line 412, correct the reference number: is it 130? Change numbering as it appears before 102.

Line 424, What is “inert bee venom”? specify.

Line 464, Typing error: dexamethasone.

Line 490, indicates the references of antitumor effects of melittin and PLA2.

Line 223, reference [49] is not related to these PLA2 activities described in the sentence.

Line 234, reference [49], is not related to any hyaluronidase activity.

[33], properly cite the reference, the article was published in the JBC.

Author Response

This review provides an overview of the history of bee venom therapy, its production and extraction as well as its application in inflammatory, infectious and cancer processes with a focus on veterinary treatments. Despite numerous reviews on the subject, it is worth publishing this article for its original focus: use of bee venom in veterinary medicine as well as interesting information about production and extraction of bee venom.

I suggest  to the authors to read Khalil et al 2021 (Bee Venom: From Venom to Drug. Molecules 2021, 26, 4941) an interesting and recent review, which presents an important and recent bibliographic survey of the subject, focusing on pharmacological activities.

References are from peer-reviewed journals, however I am not familiar with specific beekeeping publications.

Response

We thank the reviewer for his advice and comments that enable us to improve the overall quality of the manuscript. An extensive rewrite of the work was carried out in accordance with the reviewers' suggestions. Minor corrections highlighted by you have been made.

General comments

Write the names of the bacteria in italics.

Response (R): corrected as suggested

Normalize the writing of PLA2 in the text.

R.: Done

Specific comments

Line 90: indicates the references for the diseases described in the paragraph.

R.: now added

Line 157: repetition of sentences in the paragraph.

R.: deleted

L 165, Typing error: “carboxy-terminal region”.

R.: corrected as suggested

L183, Typing error: apamin  is a neurotoxin.

R.: corrected as suggested

Line 263, indicates the reference of comparison of extraction method.

R.: now added

Line 292, indicates the references of the antioxidant activities of toxins cited in the paragraph.

R.: now added

Line 323, the sentence seems related to the following item 5.2. Antimicrobial and antiviral activity.

R.: deleted

Line 332, indicates the reference to melittin's mechanism of action.

R.: now added

Line 412, correct the reference number: is it 130? Change numbering as it appears before 102.

R.: corrected as suggested

Line 424, What is “inert bee venom”? specify.

R.: many thanks for this comment. It is a typo that has now been corrected

Line 464, Typing error: dexamethasone.

R.: corrected as suggested

Line 490, indicates the references of antitumor effects of melittin and PLA2.

R.: now added

Line 223, reference [49] is not related to these PLA2 activities described in the sentence.

R.: corrected

Line 234, reference [49], is not related to any hyaluronidase activity.

R.: the reference has been corrected as suggested

[33], properly cite the reference, the article was published in the JBC.

R.: corrected as suggested

Reviewer 2 Report

1. Relatively higher similarity  with "Bee Venom: An Updating Review of Its Bioactive Molecules and Its Health Applications" (https://doi.org/10.3390/nu12113360)

2. From section one to section four, only two pages of text should be included.

3. The language of the manuscript must be improved for high clarity.

4. In each subsection of Section 5, specific examples of diseases/conditions for which this alternative approach should be used must be provided.

5. Check the attached document for feedback

6. In its current form, it appears to be a topic overview rather than a review.

Author Response

  1. Relatively higher similarity  with "Bee Venom: An Updating Review of Its Bioactive Molecules and Its Health Applications" (https://doi.org/10.3390/nu12113360)

R.: we thank the reviewer for his important suggestions and comments. The manuscript has been extensively revised and new information has been added in accordance with the advice given.

  1. From section one to section four, only two pages of text should be included.

R.: as the first reviewer pointed out, this section is particularly important and provides important introductory information on the subject. We therefore prefer to keep it with your approval.

  1. The language of the manuscript must be improved for high clarity

R.: many sentences and the English language in general have been revised to allow for better understanding by those who will read the manuscript.

  1. In each subsection of Section 5, specific examples of diseases/conditions for which this alternative approach should be used must be provided.

R.: thank you very much for this comment which helps to improve the manuscript. Several published studies involving murine models and animals have been included as suggested.

  1. Check the attached document for feedback

R.: all attache advices have now been included in the manuscript

  1. In its current form, it appears to be a topic overview rather than a review.

R.: we have added more details that we believe have changed the appearance of the manuscript.

Reviewer 3 Report

This manuscript requires some English language copyediting.

Overall, the manuscript is well written but major concern is that it's topic and content are not geared towards veterinary sciences.

Author Response

This manuscript requires some English language copyediting.

R.: we thank the reviewer for this comment. An extensive review of the English language was carried out.

Overall, the manuscript is well written but major concern is that it's topic and content are not geared towards veterinary sciences.

R.: we thank the reviewer for this observation. The manuscript has been implemented in its entirety and several animal studies have been added in accordance with the advice.